# Role of Oxylipins in the Inflammatory-Related Diseases NAFLD, Obesity, and Type 2 Diabetes

**DOI:** 10.3390/metabo12121238

**Published:** 2022-12-09

**Authors:** Mariya Misheva, Jethro Johnson, James McCullagh

**Affiliations:** 1Chemistry Research Laboratory, University of Oxford, 12 Mansfield Road, Oxford OX1 3TA, UK; 2Oxford Centre for Microbiome Studies, Kennedy Institute of Rheumatology, Roosevelt Drive, Headington, Oxford OX3 7FY, UK

**Keywords:** eicosanoid, inflammation, diagnosis, progression, treatment, mass spectrometry

## Abstract

Oxygenated polyunsaturated fatty acids (oxylipins) are bioactive molecules established as important mediators during inflammation. Different classes of oxylipins have been found to have opposite effects, e.g., pro-inflammatory prostaglandins and anti-inflammatory resolvins. Production of the different classes of oxylipins occurs during distinct stages of development and resolution of inflammation. Chronic inflammation is involved in the progression of many pathophysiological conditions and diseases such as non-alcoholic fatty liver disease, insulin resistance, diabetes, and obesity. Determining oxylipin profiles before, during, and after inflammatory-related diseases could provide clues to the onset, development, and prevention of detrimental conditions. This review focusses on recent developments in our understanding of the role of oxylipins in inflammatory disease, and outlines novel technological advancements and approaches to study their action.

## 1. Introduction

Oxylipins are bioactive lipid mediators derived from the enzymatic or non-enzymatic oxidation of polyunsaturated fatty acids (PUFAs). The enzymatic oxidation of PUFAs is carried out by cyclooxygenases (COX), lipoxygenases (LOX) [1,2,3,4] or cytochrome P450s (CYP) [5,6,7,8,9,10]. PUFAs are those fatty acids that have more than one double bond in their backbone. When the final double bond is at the carbon–carbon double bond in the n-6 position (the sixth bond counting from the methyl end), they are referred to as omega-6 fatty acids. When that bond is in the n-3 position, these are known as omega-3 fatty acids [11]. There are two PUFAs in the human body that cannot be produced endogenously, linoleic and alpha-linolenic acid (LA and ALA), and accordingly they are defined as essential fatty acids. LA is an omega-6 fatty acid that is metabolised to arachidonic acid (AA) and dihomo-gamma-linolenic acid (DGLA). ALA is an omega-3 fatty acid which is converted to eicosapentaenoic acid (EPA) and docosahexaenoic acid (DHA). Therefore, based on research discoveries so far AA, DGLA, EPA and DHA are the non-essential fatty acids which give rise to numerous oxylipin families (Figure 1). Importantly, mammals including humans cannot convert omega-6 to omega-3 PUFAs. Hence, the levels of these oxylipins reflect the consumption of omega-3 or omega-6 PUFAs. Specific omega-3 and omega-6 metabolites have been implicated to have opposing beneficial or detrimental effects in human health and disease, with omega-3-derived compounds found to be generally anti-inflammatory while oxylipins originating from omega-6 fatty acids have predominantly pro-inflammatory effect [12,13]. The pleiotropic effects of these compounds are increasingly appreciated with different oxylipins having dissimilar roles in physiological and pathophysiological processes (reviewed excellently in [14,15,16,17,18]). Importantly, the superclass of oxylipins includes hundreds of structurally and stereochemically distinct species [14]. Thus, profiling of oxylipins can be an important tool in investigating their role in pathophysiological processes. 

Although all non-essential fatty acids can serve as precursors for the synthesis of oxylipins, the eicosanoids identified to date predominantly originate from AA due to enzymatic activity (Figure 2). In addition, some eicosanoids can form as a result of non-enzymatic oxidation, although those serve mostly as markers for oxidative stress in vivo (eicosanoid biology reviewed in [19]). However, the majority of the oxylipins mentioned in this review are products of the enzymes LOX, COX, or CYP. Thus, research interest focuses mostly on the enzymatically produced oxylipins.

Prostaglandins (PG) are derived from AA by COX, and were the first oxylipins to be identified [20]. Their importance in disease has been investigated ever since it was reported that aspirin inhibits COX enzymes and the formation of their metabolites (prostaglandins) [21,22]. Since then, oxylipins have been implicated in inflammation, cardiovascular disease, atherosclerosis, and multiple other physiological and pathophysiological processes [1,5,6,7,8,9,23,24,25,26,27,28]. In September 2022, searching PubMed with ‘oxylipin’ as the keyword returned 6848 results, while ‘eicosanoids’ gave 154,900 results. Although technically eicosanoids refer to lipids containing 20 carbon atoms, with time the term ‘eicosanoids’ has expanded to also include similar metabolites of other PUFAs and therefore is also used as a synonym for ‘oxylipin’. Thus, in this publication the two terms, oxylipin and eicosanoids, will be used interchangeably.

The impressive number of entries for ‘eicosanoid’ in PubMed underscores the efforts put into determining the role of those bioactive molecules in health and disease. The sheer volume of published articles presents a substantial obstacle in reviewing all publications. Therefore, this review focusses on research articles published in the last three years (2019–2022) with the aim of highlighting the latest developments in several key areas. Exceptions to this are articles published prior to 2017 but considered to present results of major importance to the research area. The main areas covered in this review are: (1) non-alcoholic fatty liver disease (NAFLD), (2) obesity, diabetes, and autoimmune diseases, and (3) technological advancements and future directions.

## 2. Non-Alcoholic Fatty Liver Disease (NAFLD)

NAFLD is a chronic liver disease that results in an excessive increase in fat (steatosis) in the liver without evidence of other causes of liver disease (such as alcohol). There are two progressive subtypes of NAFLD: non-alcoholic fatty liver (NAFL) and non-alcoholic steatohepatosis (NASH) which is progression of NAFLD beyond NAFL alone [29]. The main difference between the two types is the presence of inflammation. NAFL is characterised by little or no inflammation or liver damage. NAFL usually does not cause complications, though it can lead to pain from liver enlargement [30,31]. In comparison, NASH is categorised by inflammation and liver damage in addition to the build-up of fat. NASH can result in the development of cirrhosis which in turn can lead to liver cancer [32,33,34,35]. Approximately 25% of the general population will develop NAFLD and, of those, 1.5–6.45% will progress to NASH [34,36]. Thus, detection, prevention, and treatment of NAFLD is a major challenge for modern medicine. 

Although the pathogenesis of NAFLD is still not completely understood, its connection to fat build-up is clear. Bodyweight reduction among NAFLD patients showed lower levels of triglycerides, cholesterol, and the eicosanoids 5-Oxo-eicosatetraenoic acid (5-oxo-ETE) and lipoxinA4 (LXA4), and was linked to a significant improvement in steatosis [37]. Analysis of liver tissue samples from obese patients has indicated a complicated relationship between eicosanoid levels and paediatric NAFLD. CYP-produced eicosanoids have been shown to be downregulated during fibrosis, but increased in the steatosis stage, while the levels of precursor PUFAs and the production of hydroxyeicosatetraenoic acids (HETEs), hydroxyeicosapentaenoic acids (HEPEs), and hydroxydocosahexaenoic acids (HDHAs) remained unchanged [38]. CYP-derived oxylipins are now investigated for their potential hepatoprotective effects. Quantification of circulating epoxyeicosatrienoic acids (EETs) has indicated that when compared with controls, total EET; 11,12-EET; 14,15-EET, total dihydroxyeicosatrienoic acids (DHET); 11,12-DHET; and 14,15-DHET were significantly lower in NAFLD and, furthermore, total EET and DHET were lower in NASH compared with steatosis (i.e., NAFL, but not NASH) [39]. EETs are derived from AA, therefore they are omega-6 fatty acids, yet they appear to be potential targets for prevention of NASH development. Indeed, in an animal study it was reported that receiving a diet enriched in the omega-6 precursor LA during pregnancy, lactation and growth did not induce obesity in offspring as previous reports had suggested. The authors reported that high LA intake leads to the simultaneous synthesis of both pro-inflammatory (5-HETE, 12-HETE, Leukotriene B4 (LTB4)) and anti-inflammatory (8,9-EET) and pro-resolving (LXA4) bioactive molecules, thus highlighting the need for a detailed assessment of individual eicosanoids in the development of hepatic steatosis [40].

NAFLD is usually asymptomatic and diagnosed after the incidental finding of abnormal liver disease or steatosis during liver imaging, and after other causes of liver disease have been excluded [41]. Liver biopsy is the gold standard for diagnosing and staging NAFLD (reviewed in more detail in [42,43]). Therefore, there is significant interest in developing non-invasive and accurate methods for NAFLD diagnosis and assessing its progression. Measuring oxylipin levels presents one possible approach. Evaluation of the levels of eicosanoid concentration in serum and liver tissue in rats during steatosis indicated a strong positive correlation between 9-HODE and 13-HODE and NAFLD progression. In addition, the study reported a moderate correlation between the levels of the specialised pro-resolving mediator (SPM) resolvin E1 in liver and serum. The levels of HETE also showed association with the NAFLD phenotype, but the link was weak [44]. Thus, there is potential for eicosanoid profiles to be used as biomarkers for tissue remodelling that leads to NAFLD.

Considering the negative impact of NAFLD on human health, much research has also focused on effective treatment approaches in addition to disease diagnosis and progression. One study evaluated the possible use of fibrates (a class of amphipathic carboxylic acids) and omega-3 PUFAs, typically used to combat hypertriglyceridemia. The results indicated a reduction in plasma concentration of the pro-inflammatory prostaglandin E2 (PGE2), as well as prostaglandin E1 (PGE1), prostaglandin D1 (PGD1) and thromboxane B2 (TXB2) (omega-3 PUFAs) and reduced CYP-derived DHETs (fibrate), but increased prostacyclin, 17-HDHA, 18-HEPE, dihydroxydocosapentaenoic acids (19,20-DiHDPA)(omega-3 PUFAs). Thus, omega-3 PUFAs and fibrate have some effect on the formation of lipid mediators with potential effects on chronic inflammation, but their use for treating NAFLD needs further investigation [45]. 

Lipidomics analysis of blood and liver mouse samples indicated that *Sagittaria sagittifolia* (SSG), a flowering plant in the family Alismataceae, has protective effects against high-fat diet-induced NAFLD. SSG interfered with AA metabolism via the Nrf/HO-1 signalling pathway during oxidative stress in the liver [46]. Another study indicated that the sodium–glucose cotransporter 2 inhibitor canagliflozin mitigates fatty liver and hyperglycemia without affecting body weight in mice. Lipidomics analysis showed that canagliflozin increased the levels of PGE2 and resolvin E3 in the mouse liver. Interestingly, PGE2 reduced fat depositions in mouse primary hepatocytes exposed to palmitic acid [47]. Thus, certain eicosanoids might be used as potential treatment options to tackle NAFLD progression.

## 3. Obesity and Diabetes

Apart from NAFLD, chronic inflammation is associated with obesity which in turn contributes to the development of insulin resistance and type 2 diabetes (T2D). The plasma level of three eicosanoids (unknown eicosanoid (EIC 62), 8-iso-prostaglandin A1 (8-iso-PGA1), and 12-hydroxy-5,8,10-heptadecatrienoic acid (12-HHTrE) was shown to predict incident T2D [48] and the resolvin D2/LTB4 ratio, and may serve as a biomarker of prognosis for ischemic stroke [49]. An analysis of clinical samples revealed significant differences in the levels of four oxylipins (PGF2α, PGE2, 15-keto-PGE2, and 13,14-dihydro-15-keto-PGE2) between T2D patients and the corresponding lean and obese control subjects, with the combination of PGF2α and 15-keto-PGE2 having the most predictive value [50]. Targeted lipidomics analysis of human urine samples indicated that metabolite products of PGD and PGE are associated with low-grade chronic inflammation in obesity [51]. Upregulated PGE2 production by β-cells may have a role in the β-cells’ adaptation response to obesity and insulin resistance in T2D when PGE2 and its receptor EP3 are highly expressed [52]. Further supporting the connection between increased oxylipin levels, insulin resistance and T2D, is how weight-loss intervention in obese, insulin-resistant, sedentary women was accompanied by decreased concentrations of 9,10-DiHODE, 12,13-DiHODE and 9,10-DiHOME [53], while 5-LOX leukotrienes were increased [54]. Thus, eicosanoids have shown potential as predictors of obesity and T2D.

However, the levels of other oxylipins such as the SPMs and hydroxy-DHA metabolites were shown to be lower in obesity and white adipose tissue inflammation [55]. Animal studies revealed that increasing the bioavailability of SPMs such as Maresin-1 minimises inflammation and mediates therapeutic actions [56]. It is not surprising therefore that eicosanoids are seen as potential targets to combat obesity. One such approach suggests seeking ways to increase the energy expenditure of thermogenic tissues such as brown and brite adipose tissue. Both 5-HETE and 5,6-EET have been reported to be consistently associated with the abundance of those tissues, though further studies are needed to determine whether those eicosanoids are candidates that affect thermogenic capacity [57]. In addition, data has suggested that PGE2 metabolites could be an indicator of the efficacy of mesenchymal stem cell treatment in systemic lupus erythematosus. However, further studies and clinical trials are needed [58]. Thus, eicosanoids also present viable therapeutic targets in a range of pathophysiological conditions.

Another potential approach to influence eicosanoid levels could be through diet and its effect on gut microbiota composition. Recently, the role of eicosanoids as modulators of inflammation in the gastrointestinal tract has been reviewed [59] as well as changes in gut microbiota composition caused by dietary and endogenous lipids, including eicosanoids [60]. These reviews describe very well the link between diet, gut microbiota, and eicosanoids. Therefore, here we will mention only a few examples of that association. Animal studies have demonstrated that gut microbiota dysbiosis modifies the oxylipin profile in healthy and obese rats, with several significant correlations between different bacteria taxa and eicosanoids. Among those, the positive association between Proteobacteria and LTB4 was found to be especially strong [61]. In addition, gut bacteria have been proposed to be responsible for the post-prandial decrease in soluble epoxide hydrolase, an enzyme responsible for metabolising EETs to the less active diols [62]. Moreover, prostaglandin E2 (PGE2), a well-known mediator of inflammation, was found to inhibit mucosal regulatory T cells in a manner regulated by gut microbiota [63]. Thus, evidence suggests that modifying nutritional intake could present a possible non-invasive approach to affect inflammation-related diseases by modulating eicosanoid production.

## 4. Technological Advancements and Future Directions

The importance of oxylipins in physiological and pathophysiological processes is underscored by their role in the ability of cells to acquire different functional phenotypes depending on the microenvironment. Omega-3 and omega-6 PUFA-derived oxylipins can modulate the inflammatory phenotype of cells [64] by acting as ligands for receptors such as the peroxisome proliferator-activated receptor PPAR [65] and prostaglandin E2 (PGE2) receptor PTGER4 [66]. It is not surprising then that there are continuous efforts to improve their detection and quantitation. In this section, we will outline current and new technical approaches to determine oxylipin profiles.

The eicosanoid class comprises hundreds of structurally and stereochemically distinct species derived either enzymatically or non-enzymatically from a handful of precursors. It is therefore not surprising that within the group of eicosanoids can be found isomers (e.g., PGE2 and PGF2α) with different biological functions [14]. In addition to their diversity, eicosanoids occupy a small mass range and are found in low nanomolar concentrations in biological samples (e.g., human plasma and murine bone marrow-derived macrophages [67]). Thus, their detection and evaluation require methods that are sensitive, selective, and reproducible.

Currently, mass spectrometry (MS) is the main technique used to interrogate the lipidome as demonstrated by the methods used in the experimental articles on NAFLD, obesity and diabetes reviewed here (Table 1). Mass spectrometry is a powerful technique for identifying and quantifying known and unknown analytes, and offers high specificity and selectivity. Technical advances have given mass spectrometry a range of tools to obtain information at the molecular level from samples, such as determining the molecular weight of an analyte, its separation from isomeric and isobaric species, its chemical formula, its molecular structure, and structural information [68,69,70]. Mass spectrometry can be used to investigate a wide range of classes of molecules, including those with small and large molecular weights, volatile and non-volatile, polar and non-polar [71,72,73].

Applications using mass spectrometry are either targeted or untargeted. Untargeted approaches attempt to measure as many compounds as possible in a sample, and can be used to discover unanticipated changes between experimental groups. This information can then be used to generate and test hypotheses, for example by applying targeted approaches which offer a selection of the best possible conditions for the detection, identification, and (absolute or relative) quantitation of particular analytes of choice, and is usually used in follow-up experiments. Targeted approaches may use multiple reaction monitoring (MRM) methods which use the knowledge of the mass of the ionised analyte and its corresponding fragments formed during the mass spectrometry analysis [74,75]. This provides a high level of selectivity and specificity, ensuring the correct analyte is targeted. It is important to note that targeted approaches do not provide global coverage, and that compounds present in the sample that are not the targets of interest will not be detected. 

For lipidomics, liquid chromatography-mass spectrometry (LC-MS) is commonly used. LC separates molecules depending on their hydrophobicity, molecular size, and polarity, and covers a broad range of non-polar and weakly polar analytes [76,77]. Another separation method that is gaining popularity in metabolomics is ion mobility coupled with mass spectrometry (IMS-MS; IM-MS) (IMS) [67,78,79,80,81,82,83]. Coupling IM to MS offers several benefits such as improved selectivity and therefore increased capability to analyse complex mixtures, and the resolving of isomeric [84] and isobaric [85] compounds. Furthermore, IMS provides additional information about the shape and conformation of the ions under defined experimental conditions [86,87], thus adding molecular, structural, and conformational information which further improves the accurate determination of metabolites in complex structures [88,89,90]. For example, IM-MS has been used to identify unknown oxylipins in different biologically relevant matrices [91]. However, like the other methods, IMS has its disadvantages such as the need for improved ion mobility resolutions.

Recently, supercritical fluid chromatography (SFC) has gained traction as an alternative technique to LC in lipidomics due to its high efficiency [92,93]. Briefly, SFC uses supercritical fluid such as CO_2_ as a mobile phase [94], which results in high separation efficiency and short separation time [95,96]. SFC-MS/MS methods have been reported to detect inflammation-related lipids including oxylipins in rats [97], as well as for the simultaneous measurement of five AA-derived metabolites (PGD2, PGE2, PGF2α, 6KetoPGF1α and LTB4) in biological samples [98]. Thus, SFC coupled to MS is a promising approach for the interrogation of the lipidome. 

Novel ionisation approaches have appeared with the development of ambient mass spectrometry, characterised by direct sampling and ionisation of the analytes with no or minimal sample preparation [99]. Ambient ionisation MS has been successfully employed to rapidly differentiate bacterial species based on their lipid profiles [100]. Mass spectrometry imaging (MSI) developments allow more detailed investigations of biological questions such as the biochemical origin of lipid spatial distribution (reviewed in [101]). Examples of MSI techniques include secondary ion mass spectrometry (SIMS), Desorption electrospray ionisation (DESI) and matrix-assisted desorption/ionisation (MALDI). Lipid characterisation is now possible due to technological advances in SIMS (reviewed in [102]). DESI-MSI can record 2D distributions of polar lipids in tissue slices at ambient conditions (at atmospheric pressure) [103], and has been developed for the simultaneous imaging of polar and non-polar lipids in mouse brain tissue [104]. Improvements in MALDI resolution resulted in the identification and localisation of lipids within the kidney, as well as the localisation of lipid droplets with lesion-specific macrophages [105]. Thus, further method developments could provide the means to image eicosanoids in a variety of biological samples.

## 5. Conclusions

In recent years there has been substantial research focused on eicosanoids, which started with the discovery of prostaglandins and elucidating their role in inflammation. Continuous efforts have resulted in the identification of other oxylipin classes, with resolvins being the latest addition to that list. Eicosanoids have been found to have both pro- and anti-inflammatory roles, and have been shown to be associated with specific disease manifestations and, therefore, could potentially serve as biomarkers [50,106]. In addition, eicosanoids have been found to have a role in the host–pathogen interactions, thus presenting a prospective therapeutic target in bacterial infections [107,108,109]. It is not surprising then that the field of lipidomics keeps expanding with the application of novel techniques and approaches such as SFC, ambient MS, and ion mobility increasing not only our knowledge of known eicosanoids but also leading to the identification and characterisation of novel lipids (including oxylipins) and other lipid classes. 

## Figures and Tables

**Figure 1 metabolites-12-01238-f001:**
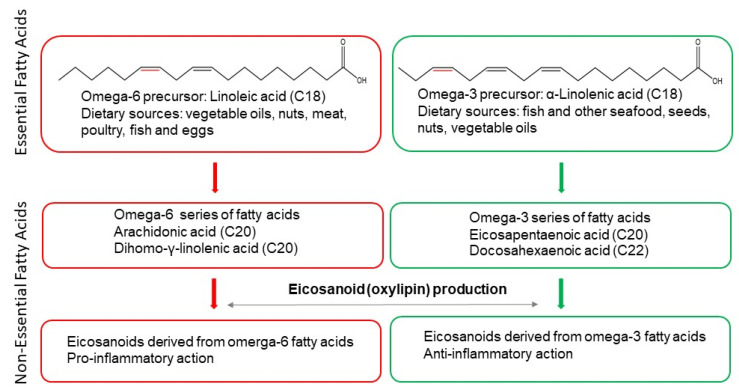
General overview of essential and non-essential polyunsaturated fatty acids. LA is a non-essential fatty acid and precursor of the omega-6 series of fatty acids and eicosanoids which have mostly pro-inflammatory action. ALA is a non-essential fatty acid and precursor of the omega-3 series of fatty acids and eicosanoids which have predominantly anti-inflammatory effect. Omega-6 and omega-3 double bonds are marked in red. Chemical formulas were created using ChemDraw^®^. More detailed description of the role of PUFAs in inflammatory processes, including exceptions, is outlined in the main text.

**Figure 2 metabolites-12-01238-f002:**
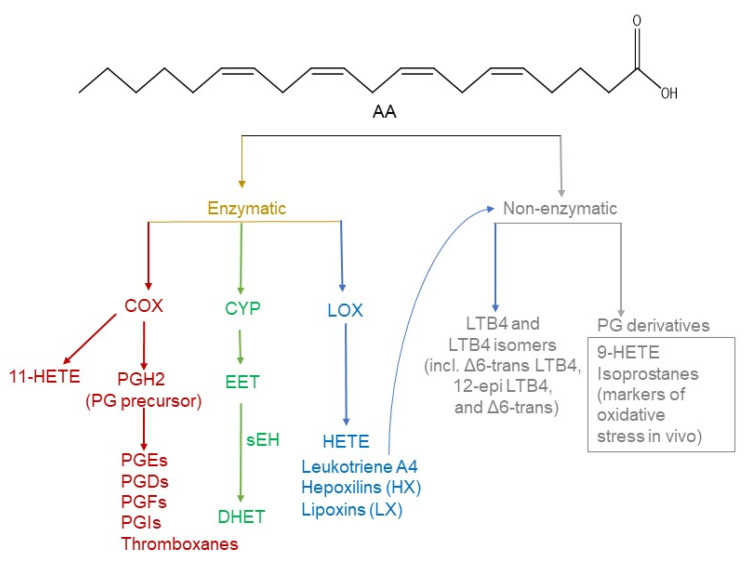
Arachidonic acid is the main precursor of eicosanoids. Eicosanoids can be produced enzymatically by COX, LOX, and CYP, and non-enzymatically. COX-derived eicosanoids include prostaglandins as well as 11-hydroxyeicosatetraenoic acid (HETE). CYP enzymatic activity produces EETs that are further hydrolysed to DHET by soluble epoxide hydrolase (sEH). HETEs, leukotrienes, hepoxilins, and lipoxins are produced by LOX. Leukotriene A4 is subjected to further non-enzymatic hydrolysis to form Leukotriene B4 LTB4 and its derivatives. Non-enzymatic oxidation of AA results in the formation of eicosanoids used as markers of oxidative stress in vivo. Chemical formula for AA was created using ChemDraw^®^.

**Table 1 metabolites-12-01238-t001:** Methods of eicosanoid detection.

Oxylipin Abbreviation	Full Name	Fatty Acid Source	Enzyme	Method of Detection	REF
5-HETE	5-Hydroxyeicosatetraenoic acid	AA	LOX	Liquid chromatography-mass spectrometry (LC-MS/MS) Unspecified lipidomics approach	[40] [57]
12-HETE	12-Hydroxyeicosatetraenoic acid	AA	LOX	LC-MS/MS	[40]
18-HEPE	18-Hydroxyeicosapentaenoic acid	EPA	unknown, non-enzymatic	UPLC-MS/MS	[45]
17-HDOHE (17-HDHA)	17-Hydroxydocosahexaenoic acid	DHA	LOX	UPLC-MS/MS	[45]
9-HODE	9-Hydroxyoctadecadienoic acid	LA	LOX	HPLC	[44]
13-HODE	13-Hydroxyoctadecadienoic acid	LA	LOX	HPLC	[44]
5-OxoETE	5-Oxo-eicosatetraenoic acid	AA	LOX	HPLC	[37]
9,10-DiHOME	9,10-Dihydroxyoctadecenoic acid	LA	CYP	LC-MS/MS	[53]
11,12-DiHETrE (DHET)	11,12-Dihydroxyeicosatrienoic acid	AA	CYP	LC-MS	[39]
14,15-DiHETrE (DHET)	14,15-Dihydroxyeicosatrienoic acid	AA	CYP	LC-MS	[39]
RvD2	7S,16R,17S-Trihydroxydocosahexaenoic acid	DHA	LOX	Enzyme immunoassay (EIA)	[49]
LTB4 (Leukotriene B4)	5S,12R-Dihydroxyeicosatetraenoic acid	AA	LOX	LC-MS/MS Enzyme immunoassay (EIA) LC-MS/MS	[40] [49] [61]
LXA4 (Lipoxin A4)	5S,6R,15S-Trihydroxyeicosatetraenoic acid	AA	LOX	HPLC LC-MS/MS	[37] [40]
Maresin1	7R,14S-Dihydroxydocosahexaenoic acid	DHA	LOX	LC-MS/MS	[56]
5,6-EET (EpETrE)	5(6)-Epoxyeicosatrienoic acid	AA	CYP	Unspecified lipidomics approach	[57]
8(9)-EET (EpETrE)	8(9)-Epoxyeicosatrienoic acid	AA	CYP	LC-MS/MS	[40]
11(12)-EET (EpETrE)	11(12)-Epoxyeicosatrienoic acid	AA	CYP	LC-MS	[39]
14(15)-EET (EpETrE)	5(6)-Epoxyeicosatrienoic acid	AA	CYP	LC-MS	[39]
PGD1 (prostaglandin D1)	9α,15S-dihydroxy-11-oxo-prost-13E-en-1-oic acid	DGLA	COX	UPLC-MS/MS	[45]
PGE1 (prostaglandin E1)	9-oxo-11α,15S-dihydroxy-prost-13E-en-1-oic acid	DGLA	COX	UPLC-MS/MS	[45]
PGE2 (Prostaglandin E2)	9-oxo-11α,15S-dihydroxy-prosta-5Z,13E-dien-1-oic acid	AA	COX	UPLC-MS/MS LC-MS/MS UPLC-MS/MS Direct injection MS ELISA LC-MS/MS	[45] [47] [50] [52] [58] [63]
13,14-dihydro-15-keto PGE2	9,15-dioxo-11α-hydroxy-prost-5Z-en-1-oic acid	AA	COX	UPLC-MS/MS	[50]
PGF2α	9α,11α,15S-trihydroxy-prosta-5Z,13E-dien-1-oic acid	AA	COX	UPLC-MS/MS	[50]
TXB2 (Thromboxane B2)	9α,11,15S-trihydroxythromba-5Z,13E-dien-1-oic acid	AA	COX	UPLC-MS/MS	[45]
19,20-DiHDPA	19,20-dihydroxy-4Z,7Z,10Z,13Z,16Z-docosapentaenoic acid	DHA	CYP	UPLC-MS/MS	[45]
8-iso-prostaglandin A1	9-oxo-15S-hydroxy-(8β)-prosta-10,13E-dien-1-oic acid	DGLA	non-cyclooxygenase origin (minor impurity during commercial preparations of PGE1)	LC-MS	[48]
12-HHTrE	12S-hydroxy-5Z,8E,10E-heptadecatrienoic acid	AA	COX	LC-MS	[48]
15-keto-PGE2	9,15-dioxo-11α-hydroxy-prosta-5Z,13E-dien-1-oic acid	AA	COX	UPLC-MS/MS	[50]
9,10-DiHODE	9,10-dihydroxy-12Z,15Z-octadecadienoic acid	LA	CYP	LC-MS/MS	[53]
12,13-DiHODE	12,13-dihydroxy-9Z,15Z-octadecadienoic acid	LA	CYP	LC-MS/MS	[53]
RvE3	17R,18S-dihydroxy-5Z,8Z,11Z,13E,15E-eicosapentaenoic acid	EPA	LOX	LC-MS/MS	[47]

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
