# Peer review of "Role of Oxylipins in the Inflammatory-Related Diseases NAFLD, Obesity, and Type 2 Diabetes"

_metabolites, 2022, doi:10.3390/metabo12121238_

Round 1

Reviewer 1 Report

The authors have done a great job setting out the review's scope at the beginning, followed by detailed descriptions and citations of every aspect covered in the paragraphs. The authors have also provided figures and tables to summarize their points effectively.

The authors Misheva et al., have submitted a review article entitled “Role of oxylipins in the progression of inflammatory-related diseases”. In the review the authors have described the different types of oxylipins and discussed their roles in a few inflammatory related diseases. The authors have started the review with a detailed introduction on the classification of different oxylipins and described the pathways involved in the biosynthesis of these molecules. The authors have also described the dietary sources of
these molecules. Further, the authors narrowed down on the oxylipins produced from arachidonic acid, which is a type of omega-6 fatty acid. The authors have followed this with detailed discussion on the levels
of different molecules mostly derived from arachidonic acid in different diseases like non-alcoholic fatty liver diseases (NAFLD), obesity and diabetes. In the final section the authors have also discussed the techniques and technological advances in the measurement of these molecules in different samples. Overall, the authors have covered quite a lot of research in the field. Reworking the review on the following lines could help in accurately depicting its relevance –
Major comments:
While the authors have only discussed the levels and modulation of different eicosanoids derived from arachidonic acid, the title and abstract of the review projects that the review covers all the other types of oxylipins of omega-6 fatty acids (where arachidonic acid belongs) and omega-3
fatty acids. However, as noted by the authors that most of the available research on eicosanoids is only carried out on arachidonic acid derivatives. Therefore, the review could only cover this topic and more research on other known derivatives of other eicosanoids is warranted to provide a complete picture of the field. It would be helpful for the authors to state this.
The authors have focused only on NAFLD, obesity and diabetes as the inflammatory related diseases. Since, there are other inflammatory related diseases associated with the body, e.g. atherosclerosis, it would be helpful for the authors to revise the title and the manuscript to depict that the review focuses on these diseases instead of a more general title and scope.
The discussion about the techniques and technological advancements in the measurement of these eicosanoids have not been discussed and incorporated in the title and the abstract. It would be helpful to include these details in the aforementioned sections.

Reviewer 2 Report

This is an interesting review on the role of lipid mediators derived from polyunsaturated fatty acids. They are involved in many biological processes and are therefore of growing interest. 

Comments:

1. Figure 1 shows that omega-6 PUFAs are a source of proinflammatory mediators. However, arachidonic acid is a substrate for proresolving lipoxins. There are further mentions of this.

2. In the first part of the article, it would have been helpful to add general information about the basic biological properties of oxylipins. This would have helped improve Figure 1.

3. In the main part of the article, it is recommended to make the sections more consistent in order to present an understanding of the links between biological actions and the role of oxylipins in disease pathogenesis. 

4. Table 1 lists either enzymes, such as 5-LOX, or just the name of an enzyme group, such as LOX. Does this mean that all lipoxygenases are involved in fatty acid fermentation to this product or is there another interpretation as to why a specific enzyme is not listed? 

5. Section 4 would be useful to expand on the known problems of oxylipins detection and the key directions to solve them.

Reviewer 3 Report

This is a well-written and well-structured manuscript. This review emphasizes the significant role of oxylipins in the progression of inflammatory-related diseases. This is the actual research topic.

I have some comments:

It is necessary to strengthen the research criteria for selecting a number of entries  with the Web of Science and Scopus databases.

Also, the Authors should emphasize that this is a narrative review.
